# Epoxy Resin-Based Materials Containing Natural Additives of Plant Origin Dedicated to Rail Transport

**DOI:** 10.3390/ma16227080

**Published:** 2023-11-08

**Authors:** Anna Masek, Olga Olejnik, Leszek Czechowski, Filip Kaźmierczyk, Sebastian Miszczak, Aleksandra Węgier, Sławomir Krauze

**Affiliations:** 1Institute of Polymer and Dye Technology, Lodz University of Technology, Stefanowskiego Str. 16, 90-537 Lodz, Poland; olga.olejnik@dokt.p.lodz.pl (O.O.); aleksandra.wegier@dokt.p.lodz.pl (A.W.); 2Department of Strength of Materials, Lodz University of Technology, Stefanowskiego Str. 1/15, 90-537 Lodz, Poland; leszek.czechowski@p.lodz.pl (L.C.); filip.kazmierczyk@dokt.p.lodz.pl (F.K.); 3Institute of Materials Science and Engineering, Faculty of Mechanical Engineering, Lodz University of Technology, Stefanowskiego Str. 1/15, 90-537 Lodz, Poland; sebastian.miszczak@p.lodz.pl; 4S.Z.T.K. “TAPS”—Maciej Kowalski, ul. Borowa 4, 94-247 Lodz, Poland; slawomir.krauze@taps.com.pl

**Keywords:** epoxy resin, quercetin, starch, biocomposite, pro-ecological, rail transport

## Abstract

The presented study is focused on the modification of commercially available epoxy resin with flame retardants by means of using natural substances, including quercetin hydrate and potato starch. The main aim was to obtain environmentally friendly material dedicated to rail transport that is resistant to the aging process during exploitation but also more prone to biodegradation in environmental conditions after usage. Starch is a natural biopolymer that can be applied as a pro-ecological filler, which may contribute to degradation in environmental conditions, while quercetin hydrate is able to prevent a composite from premature degradation during exploitation. To determine the aging resistance of the prepared materials, the measurements of hardness, color, mechanical properties and surface free energy were performed before and after solar aging. To assess the mechanical properties of the composite material, one-directional tensile tests were performed for three directions (0, 90, 45 degrees referred to the plate edges). Moreover, the FT-IR spectra of pristine and aged materials were obtained to observe the changes in chemical structure. Furthermore, thermogravimetric analysis was conducted to achieve information about the impact of natural substances on the thermal resistance of the achieved composites.

## 1. Introduction

Growing environmental awareness and public interest, new environmental regulations and the unsustainable consumption of petroleum resources have led to the search for new materials that are more environmentally friendly [1,2]. Therefore, many biopolymers and natural substances are promising compounds to replace conventional polymers and their additives that are derived from non-renewable resources to obtain biocomposites, which are more pro-ecological materials [3,4]. For instance, biofillers, including starch, are highly valued for their natural origin and abundance in nature, biodegradability as well as low cost. However, many prepared biocomposites are characterized by some drawbacks, including moisture absorption characteristics, poor heat resistance and low dimensional stability. Moreover, in some cases, due to the differences in structural properties between the natural filler and the polymer matrix, it is difficult to achieve satisfactory dispersion of the additive and, thus, a composite with the required properties [5]. The addition of biofillers to the polymer matrix may also contribute to the deterioration of the material’s resistance to ageing [6]. Therefore, some modifications of these fillers or additional substances are necessary to obtain materials with satisfactory properties [5,6].

Starch is a natural polymer derived from potatoes, rice and corn. This plant polysaccharide is not only abundant, inexpensive and biodegradable but also has good film-forming properties [7]. The structure of starch contains two distinct components. The first one is a water-soluble amylose that is characterized by a linear unbranched chain with α-1-4-glycosidic linkages, and the second one is amylopectin with a branched structure that consists of α-D-glucose units joined through α-1-6-bonds. The first component makes up approximately 30% of starch, while the second one constitutes approximately 70% of this polysaccharide. These contents may be different depending on the plant source [8]. Starch is widely applied in a variety of sectors from water treatment, the food industry, drug delivery and the pharmaceutical industry to the preparation of composites [9]. In polymer technology, this biopolymer is a relevant substrate for biodegradable thermoplastic preparation [10]. The effect of adding starch or its modified forms into a polymer matrix as a biofiller was also studied, for instance, in natural rubber [11,12] and propylene [13,14]. In spite of the fact that starch may contribute to the deterioration of mechanical properties, it can also enable the biodegradation of materials [13,14].

Next to biofillers, the incorporation of natural stabilizers as well as bio-based modifiers may also lead to obtaining biocomposites with desired properties. For instance, quercetin, which is a versatile polyphenolic compound of plant origin, has become an attractive additive dedicated to polymer materials with special regards to packaging materials [15,16,17]. This flavonoid, which is of particular interest for its antibacterial and antioxidant activities [18], may be mostly found in onions, tea as well as apples [19]. The structure of quercetin contains five hydroxyl groups linked to the phenols, which are responsible for the stabilizing effect in the polymer matrix [20]. Interestingly, the abovementioned hydroxyl moieties may also participate in the curing process of materials with appropriate functional groups, for instance, epoxy ones [21,22,23]. However, this polyphenol is more famous for its antioxidant properties, which are favorable not only in polymer technology but mostly in clinical applications [24,25]. First of all, quercetin is a powerful radical scavenger of oxygen, nitric oxide radical, peroxynitrite and other reactive oxygen species (ROS), offering protection from the oxidation process [17]. Furthermore, this natural antioxidant is able to reduce metal ions, including the iron ion, which contributes to the stabilizing effect even in the polymer matrix [20]. On the other hand, quercetin is also able to absorb UV light, including UVA (λ_max_ = 265 nm) and UVC (λ_max_ = 256 nm) rays; thus, it may inhibit the photodegradation process of the material [26,27,28,29]. Moreover, the DSC and TGA results of polypropylene with quercetin proved that the addition of 0.25% of this natural polyphenol caused polymer degradation at a higher temperature (T = 301.1 °C) in comparison with that of pristine material (T = 265.9 °C). Furthermore, the incorporation of this flavonoid into the polypropylene matrix resulted in an extension of the oxidative induction time (OIT) from 8 min to 94 min at a high temperature of T = 210 °C. Therefore, quercetin may also act as a thermal stabilizer [26]. The combination of polyphenol with biofiller may prevent the premature degradation of biocomposites during exploitation. However, to achieve a better stabilizing effect, quercetin should not be chemically bonded to the biofiller surface, and only physical interactions are favored. This has been studied and demonstrated in the case of the quercetin@cellulose system incorporated into an ethylene–norbornene copolymer matrix by Cichosz et al. [6].

There is a great need for novel pro-ecological materials in different sectors, including the railway industry. For example, increased attention is being paid to the development of novel biocomposites for seats, which should be lightweight and durable [30]. In modern railways, the heavy and expensive metal parts of seats have been replaced with polymer composites, including materials made of epoxy resin, with desirable mechanical properties [31]. However, epoxy resin-based materials, because of their low resistance to aging processes such as thermal, UV and natural aging [32], require different modifications and the incorporation of various stabilizers. Both natural and synthetic stabilizers that are dedicated to epoxy resin have been extensively studied [32,33,34]. However, it must be noted that novel materials should comply with the rules of Green Chemistry. Therefore, many environmentally friendly modification methods of widely used conventional epoxy resin are carried out. On the other hand, the creation of novel, fully bio-based composites that are based on resins derived from renewable resources is also a good way to extend sustainability [35,36]. In this research, we are focused on achieving biocomposites that are dedicated to railway seats by modifying non-sustainable epoxy resin using natural additives, including starch and quercetin.

## 2. Materials and Methods

### 2.1. Reagents

The object of this research was a commercially available epoxy resin with flame retardants under the trade name of NEMresin 1011 and manufactured by New Era Materials (Modlniczka, Poland). According to the manufacturer, the mixture consisted of approximately 73 wt.% pure epoxy resin, less than 18 wt.% poly (ammonium phosphate) as well as approximately 9 wt.% graphite. Moreover, plasticization occurs at a temperature above 65 °C. The main reinforcing material used to obtain the composites was glass-woven roving fabric (2/2, 350 g/m^2^) produced by Rymatex Sp. z o.o. (Rymanów, Poland). All the materials were supplied by S.Z.T.K. “TAPS”-Maciej Kowalski, Borowa 4, 94-247 Lodz, Poland. Furthermore, quercetin hydrate (≥95% of purity), obtained from Sigma Aldrich (Munich, Germany), was selected as a natural stabilizer, and potato starch Melvit^®^ (Kruki, Poland) was applied as a filler of plant origin. Both substances were natural additives.

### 2.2. Sample Preparation

The natural additives were incorporated into the epoxy resin mixture in the following proportion: 15 phr of starch and 2 phr of quercetin, where phr means parts per hundred resin with flame retardants. All the components of the mixture were first sieved to remove too large particles. Then, all the components were precisely mixed to obtain the homogeneous effect. The prepared homogeneous mixture was applied to the form, and the composite consisted of an arrangement of alternating layers of resin and glass fabric. The composites contained three layers of glass fabric and three layers of resin, taking into account that the amount of resin with flame retardants in every layer was constant and amounted to 450 g/m^2^. The composite formation consisted of three stages and was carried out in a hydraulic laboratory press by applying special steel vulcanization molds situated between the press shelves. To maintain the constant thickness of the specimens, the frame of steel 35 cm × 35 cm was applied. Moreover, silicon foil was used in the molds as spacers to prevent the adherence phenomenon. In the first stage, the composite was plasticized at the temperature of 90 °C for 5 min with a small pressure. Then, the composite was molded at the temperature of 120 °C using 1 MPa of pressure within 8 min. Finally, each obtained sample was conditioned at 136 °C for 45 min and cooled at room temperature. The composition of the prepared samples is presented in Table 1.

### 2.3. Thermogravimetric Analysis (TGA)

The impact of the natural additives on the composite’s thermal stability was studied by means of thermogravimetric analysis (TGA), where the mass loss data of the specimen as a function of rising temperature were obtained. The analysis was performed using a TGA/DSC1 device provided by Mettler Toledo^®^ (TA Instruments, Greifensee, Switzerand) with prior calibration using indium and zinc as standards. The test was carried out in a temperature range of 25–1000 °C with a heating rate of 20 °C/min under an air atmosphere at a flow rate of 60 mL/min using crucibles with a 70 μL volume made of ceramic (polycrystal aluminium oxide).

### 2.4. Flammability and Smoke Emission Test

The flammability assessment was carried out by measuring the maximum average rate of heat emission (MAHRE) detected during sample combustion. This parameter, which indicates the tendency of a fire to spread, was measured for the composite plates using a Sychta Laboratory Sp.J. (Police, Poland) cone calorimeter in accordance with ISO 5660-1:2015 [37], applying a heat flux of 50 kW/m^2^ and a distance of 25 mm from the ignition source.

Furthermore, the smoke emission was evaluated based on EN ISO 5659-2 [38] using a heat flux of 50 kW/m^2^. The measurement was performed for the composite plates characterized by 75 × 75 × 2.5 mm dimensions using a chamber, which is a product of Sychta Laboratorium Sp.J. (Poland). The most important determined parameters were the specific optical density in the first 4 min (D_s_(4)) as well as the cumulative specific optical densities in the first 4 min (VOF_4_), which play a significant role in fire safety of railway transport, according to the EN 45545-2 (R6) standard [39].

### 2.5. Solar Aging

The samples were aged using an Atlas SC340 MHG Solar Simulator climate chamber (AMETEK Inc., Berwyn, IL, USA) with a 2500 W MHG lamp. The solar radiation intensity is reported to be 1200 W/m^2^ at 100% lamp power intensity thanks to the special rare-earth halogen lamp, which gives a unique range of solar radiation (UV, Vis, IR). The solar aging test was conducted within 770 h, where the temperature amounted to 70 °C.

### 2.6. Shore C Hardness Measurement

The Shore C hardness measurement was obtained using a Shore hardness tester type C according to the PN-ISO 868 standard [40] with a pressure force of 10 N and an indenter (35 Sh and spring force 806.50 cN) (Zwick/Roell, Herefordshire, Great Britain).

### 2.7. Fourier Transform Infrared Spectroscopy (FT-IR) Absorbance Spectra Analysis

The Fourier transform infrared spectroscopy (FT-IR) absorbance spectra were obtained by applying a Thermo Scientific Nicolet 6700 FT-IR spectrometer with diamond Smart Orbit ATR sampling equipment in the range of 4000–400 cm^−1^. The number of used scans equaled 64 at a resolution of 4 cm^−1^. In this study, the structural changes in the tested materials that appeared as a result of aging processes were studied by means of the carbonyl index, which was estimated as the ratio of the peak intensity at 1725 cm^−1^ (C=O) to the reference peak at 2875 cm^−1^ (-CH_2_-).

### 2.8. Color Measurement

The optical study of the composites was performed before and after solar aging using color measurement. The color of the composites containing the selected natural additives was measured according to the PN-EN ISO 105—J01 [41] by applying a Spectrophotometer UV–VIS CM-36001 (Konica Minolta Sensing, Inc., Osaka, Japan). This device measures the signal reflected from the surface of the sample and converts it into the human impression, well known as color. The results were presented with the CIE-Lab system (L—lightness, a—red-green, b—yellow-blue). Furthermore, the color difference (ΔE), whiteness index (Wi), chroma (Cab) and hue angle (hab) values were calculated according to Equations (1)–(4). The values Δa, Δb and ΔL used in these equations were obtained as the difference of a, b and L parameters between the samples with and without natural additives or between a composite after and before the aging process.
(1)$$∆E=\sqrt{{{\left(\Delta a\right)}^{2}+\left(\Delta b\right)}^{2}+{\left(\Delta L\right)}^{2}}$$
(2)$$W_{i}=100-\sqrt{a^{2}+b^{2}+{\left(100-L\right)}^{2}}$$
(3)$$C_{ab}=\sqrt{a^{2}+b^{2}}$$
(4)$$h_{ab}\left\{\begin{array}{c} arctg\left(\frac{b}{a}\right),\;when\;a>0\;\cap b>0\\ 180{}^{\circ}+arctg\left(\frac{b}{a}\right),\;when\;(a<0\cap b>0)\cup (a<0\cap b<0)\\ 360{}^{\circ}+arctg\left(\frac{b}{a}\right),\;when\;a>0\cap b<0 \end{array}\right.$$

### 2.9. Microscopic Observation

The surfaces of the pristine and aged composites were observed using a Leica MZ6 stereoscopic microscope (Heerbrugg, Switzerland) with MultiScan 8.0 image analysis software (CSS, Warsaw, Poland) at a magnification of 50. The internal microstructures of the composites were observed on cross-sections using a Keyence VHX-950F microscope and VH-Z100R lens. The hybrid lighting mode (bright and dark field) at 200× magnification was used.

### 2.10. Contact Angle and Surface Free Energy Measurements

The contact angles (θ_C_) for the tested composites were measured before and after the aging process using an OCA 15EC goniometer from DataPhysics Instruments GmbH (Filderstadt, Germany). The instrument worked in conjunction with the SCA 20 software module. Three types of liquids, characterized by different polarity, i.e., water, diiodomethane and ethylene glycol, were used during the measurement. For each composite, a minimum of 6 CA results were achieved. The selected syringe was a Braun DS-D 1000 SF equipped with a needle with an outer diameter of OD = 0.52 mm, inner diameter of L = 0.25 mm and length of 38.10 mm. The volume of liquid drops equaled 1 μL. According to the data gained, the surface free energy (SFE) (mJ/m^2^) was estimated using the Owens–Wendt–Rabel–Kaelble (OWRK) model, where the geometric mean of the dispersive and polar components of the liquid’s surface tension ($\sigma_{L}^{d}$ and $\sigma_{L}^{p}$) and the same components of the solid’s surface energy ($\sigma_{s}^{p}$ and $\sigma_{s}^{d}$) are included and shown in Equation (5):(5)$$\sigma_{SL}=\sigma_{S}+\sigma_{L}-2\sqrt{\sigma_{s}^{d}\sigma_{L}^{d}}-2\sqrt{\sigma_{s}^{p}\sigma_{L}^{p}}$$
where
$$\begin{array}{c} \sigma_{SL}—interfacial\;tension\;between\;solid\;and\;liquid\;interface\;\left[\frac{mJ}{m^{2}}\right]\;or\;\left[\frac{mN}{m}\right]\\ \sigma_{S}—surface\;energy\;of\;solid\;\left[\frac{mJ}{m^{2}}\right]\;or\;\left[\frac{mN}{m}\right]\\ \sigma_{L}—surface\;tension\;of\;liquid\;\left[\frac{mJ}{m^{2}}\right]\;or\;\left[\frac{mN}{m}\right]\\ \sigma_{S}^{d}—dispersive\;component\;of\;solid’s\;surface\;energy\;\left[\frac{mJ}{m^{2}}\right]\;or\;\left[\frac{mN}{m}\right]\\ \sigma_{L}^{d}—dispersive\;component\;of\;liquid’s\;surface\;tension\;\left[\frac{mJ}{m^{2}}\right]\;or\;\left[\frac{mN}{m}\right]\\ \sigma_{S}^{p}—polar\;component\;of\;solid’s\;surface\;energy\;\left[\frac{mJ}{m^{2}}\right]\;or\;\left[\frac{mN}{m}\right]\\ \sigma_{L}^{p}—polar\;component\;of\;liquid’s\;surface\;tension\;\left[\frac{mJ}{m^{2}}\right]\;or\;\left[\frac{mN}{m}\right] \end{array}$$

The presented expression (Equation (5)) was substituted in the Young equation (Equation (6)), where the polar and the dispersive components of the solid’s surface energy were determined from the regression line in a suitable plot.
(6)$$\sigma_{L}cos\;\theta_{C}=\sigma_{S}-\sigma_{SL}$$

### 2.11. One-Directional Tensile Test

The tensile tests were performed on the basis of standard UNE EN ISO 527-1:2020-01 [42] by using an INSTRON testing machine. During the test, the speed of the moveable traverse of the machine was assumed to be stable and equal to 2 mm/min. As a result of the tests, the diagrams of force vs. elongation were obtained. Young’s modulus of the composite was determined by using an extensometer with a gauge length of 50 mm. The samples were cut out from the rectangular plate according to the scheme shown in Figure 1a, where the symbols mean the following: MD—main direction, PD—perpendicular direction and 45—an angle orientated to the plate edges. The dimensions of the samples amounted to the following: L = 170 mm, b1 = 40 mm, w1 = 10 mm, w2 = 20 mm and mean thickness t = 3 mm (Figure 1b).

## 3. Results

### 3.1. Thermogravimetric Analysis (TGA)

Thermogravimetric analysis helped to evaluate the effect of the natural additives’ impact, including potato starch and quercetin, on the thermal stabilization of the composites made of epoxy resin with flame retardants (Table 2 and Figure 2). Potato starch was added in order to facilitate the biodegradation effect as well as to reduce the cost of preparation. Unfortunately, as can be noticed, the incorporation of 15 phr of potato starch into the composite shifted the temperature of the 5% mass loss of the material from 327 °C to 310 °C, which means that the composite became more vulnerable to thermal degradation. On the other hand, the addition of 2 phr of quercetin to the composite with starch allowed for an improvement in this parameter and caused a change of the temperature of the 5% material’s mass loss from 310 °C to 321 °C. Quercetin, which is a well-known natural stabilizer, has been studied, among others, as a thermal stabilizer for polypropylene (PP) [26]. The authors claimed that the presence of 0.25% of this polyphenol in the PP matrix caused an improvement in the material’s resistance to thermal degradation and, therefore, switched the decomposition onset temperature of this thermoplastic material from 265.9 °C to 301.1 °C. Similar to this report, quercetin enhanced the thermal stability of the R/S15 composite, but the referential sample was still more stable. Nevertheless, from an industrial point of view, such differences do not have an influence on the application in rail transport. Furthermore, the temperatures of the most intensive mass loss of the reference sample and the R/S15/Q2 sample are similar and amount to 368.8 °C and 367.3 °C respectively.

### 3.2. Flammability and Smoke Emission Test

The flammability and smoke emission tests help to assess the potential use of materials in railway seats, which is crucial from a fire safety point of view. The obtained results concerning these studies are presented in Table 3. One of the most important parameters considered in the design of materials for railway seats is the maximum average rate of heat emission (MAHRE). The tendency of the fire to spread can be measured with this parameter, and according to the EN 45545-2 (R6) standard [39], cannot exceed 90 kW/m^2^. In spite of the fact that the addition of potato starch into the epoxy resin caused an increase in the MAHRE parameter from 48.04 kW/m^2^ to 55.71 kW/m^2^, the obtained value is still lower than 90 kW/m^2^, which determines the potential use of the designed composite as a material for railway seats. Moreover, the addition of only 2 phr of quercetin to the material containing 15 phr of potato starch resulted in a MAHRE value similar to the referential sample and amounted to 49.07 kW/m^2^.

Another important parameter is the D_s_(4) parameter, which is the specific optical density in the first 4 min of smoke emission during a fire, determined for materials dedicated to rail transport. Based on the EN 45545-2 (R6) standard [39], this parameter cannot exceed 300. The incorporation of 15 phr of potato starch into the epoxy resin caused an increase in the D_s_(4) from 82.69 to 127.36. On the other hand, the composite containing a combination of 15 phr of starch with 2 phr of quercetin was also characterized by a higher Ds(4) parameter of 144.74. Nevertheless, these values are still acceptable from a fire safety point of view.

Moreover, the VOF_4_ parameter, which is the cumulative value of the specific optical densities in the first 4 min of the composite fire behavior test, also plays an important role in the design of railway seat materials. This parameter indicates the smoke obscuration in the first 4 min of a fire and should be less than 600 min. According to the results, all tested materials are characterized by a VOF4 parameter below 600 min. The application of 15 phr of starch or a combination of 15 phr of starch with 2 phr of quercetin into the epoxy resin composite can change this parameter from 205.56 min to higher values, including 374.36 min and 361.16 min, respectively. Nevertheless, the results obtained are not too high and allow for the use of the designed composites as railway seat materials.

### 3.3. Shore C Hardness Measurement

The Shore C hardness measurement was performed to analyze the impact of the natural additives, including potato starch and quercetin, on the static mechanical properties of the epoxy resin-based composite. The resistance to the solar aging process was also investigated. According to the results (Figure 3), all the selected natural additives did not affect the hardness results, and each of the prepared composites are characterized by a Shore C hardness of approximately 89 units. It can also be noticed that there is no significant difference between the resistance to the aging process of the studied materials. In all cases, the hardness parameter decreased to 83–84 units. These results indicate that the added substances did not deteriorate this parameter nor the resistance to solar aging.

### 3.4. Fourier Transform Infrared Spectroscopy (FT-IR) Absorbance Spectra Analysis

The aging resistance of epoxy resin-based composites was also investigated using Fourier transform infrared spectroscopy. The structure of the commercial epoxy resin used (NEMresin 1011, a snap cure type epoxy resin) was not revealed by the producer. However, some characteristic moieties were detected using FT-IR spectra. For instance, the triple band visible at 3000–2800 cm^−1^ (grey highlight) is related to the -CH stretching, and the intensity of these bands changes during aging. The intensive peak noticeable at approximately 1606 cm^−1^ corresponds to the C=C stretching, which is characteristic for aromatic compounds (green highlight). The band at approximately 1505 cm^−1^ (yellow highlight) is also related to stretching vibration in the aromatic ring. On the other hand, the significant peak present at approximately 1231 cm^−1^ (blue highlight) is due to the ester moieties and/or oxirane rings that are characteristic for epoxy materials. Furthermore, the double signal visible at 1000–1050 cm^−1^ comes from -CH_2_-OH groups (orange highlight). Finally, the intense peak at approximately 825 cm^−1^ is due to the presence of C-C stretching vibrations of the epoxy ring (purple highlight). Similar moieties are present in other epoxy resin structures [30]. During the aging process, the new bands at approximately 1725 cm^−1^ appear as a result of the oxidation process, where carbonyl groups are formed. Furthermore, during aging, some changes can also occur at approximately 3430 cm^−1^, where new -OH groups are formed [32].

The spectra obtained made it possible to calculate the carbonyl index, which is a good way of assessing the aging resistance of the material’s surface. If the carbonyl index is high, it means that many carbonyl groups have appeared, while the amount of -CH moieties has decreased as a result of the oxidation process. This phenomenon is one of the symptoms of material surface degradation. According to Figure 4, the addition of 15 phr of potato starch caused an improvement in the resistance of the epoxy resin-based materials to solar ageing, where the carbonyl index decreased from 2.5 to 1.8. The incorporation of 2 phr of quercetin into the R/S15 composite resulted in the lowest carbonyl index parameter, which was 1.2.

### 3.5. Color Measurement

In general, the pure epoxy resins are known for their clear nature, but the studied commercially available epoxy resin with flame retardants (NEMresin 1011) is characterized by a dark color, which does not change significantly during aging, even when the material is useless. Some additives are able to change the color of the reference sample or sometimes modify the color during the aging process. Quercetin belongs to the natural antioxidants but is also able to act as a colorant, which is visible in Figure 5 as well as in Figure 6, where the color changing index of the composite with the combination of starch and quercetin amounted to 9.8 and is significantly higher in comparison to the epoxy resin containing only potato starch. The important change was also visible in the chroma parameter, which increased from 2.5 to 10.9 after the addition of quercetin to the R/S15 material. The potato starch caused only insignificant changes in the color sample. Additionally, relevant changes in chroma appeared in the R/S15/Q2 sample after the solar aging process. Therefore, quercetin can be applied as a kind of natural colorant and also as a natural aging indicator in the dark type of commercial resin, which informs about the material usage.

### 3.6. Microstructure

In order to assess the internal microstructure of the composites and the impact of potato starch and quercetin, observations were carried out using optical microscopy. The microstructures of the composites were examined in cross-sections. Microscopic photos taken at a magnification of 200× are shown in Figure 7.

Figure 7a shows the microstructure of the unmodified R composite whose matrix consists of pure NEMresin 1011 epoxy resin. Three layers of reinforcement in the form of plain-weave glass fabric (1) in an epoxy resin matrix are visible, forming a laminate with a thickness of approximately 1.4 mm. The epoxy matrix contains ammonium polyphosphate (APP) as a flame retardant that is dispersed in the form of a powder (2) whose spatial distribution is quite uneven. The matrix also contains a second flame-retardant additive, graphite, which is visible in the form of a few irregular flakes (3).

Figure 7b shows the microstructure of the R/S15 composite based on the NEMresin 1011 resin with the addition of 15 phr of potato starch. Between the layers of glass fabric, matrix layers with a very high proportion of powder phases (4) can be observed. The structure of these phases, consisting of APP and starch powders, is characterized by large agglomerates, which indicates the coagulating effect of the potato starch.

Figure 7c shows a microstructure photo of the R/S15/Q2 composite that, apart from the NEMresin 1011 resin and addition of 15 phr of starch, also contains 2 phr of quercetin hydrate. The microstructure characteristics of this composite are very similar to the R/S15 sample—the epoxy resin matrix contains large agglomerate areas composed of APP and starch powders. Some of these areas show slightly yellow tints (5) of the powder phases. This coloring indicates the presence of dispersed quercetin particles in these places. Despite the significant content of potato starch and quercetin as well as their tendency to coagulate with the APP particles, these additives do not significantly affect the degree of defect in the composite epoxy matrix.

### 3.7. Surface Free Energy Measurements

According to the surface free energy results (Figure 8), potato starch contributed to an increase in the polar component from 2.69 mN/m to 5.33 mN/m. This means that the composite containing 15 phr of starch is more hydrophilic, which allows for better wettability using water. Therefore, this material can be more biodegradable in environmental conditions than the referential composite. The addition of quercetin into the sample with starch resulted in a similar polar component value in comparison to the referential sample. In all conditions, the polar component increased after solar aging, and the most visible changes were observed in the referential composite. The most resistant to solar aging was the composite with starch and quercetin, as the changes in the surface free energy before and after aging were the tiniest.

### 3.8. One-Directional Tensile Test

The results of the tensile tests of the specimens examined before and after the aging process are shown in Figure 9, Figure 10 and Figure 11. The mechanical parameters were determined for three directions (MD, PD, 45). Figure 9 displays the strain expressed in % vs. normal stress in MPa for the reference sample (pure resin). The maximum stress and Young’s modulus for the samples MD were obtained as 277–283 MPa and 17.3–18.4 GPa, respectively. Taking into consideration the PD samples, these values are lower by approximately 15–20% in a comparison to the MD samples. Based on the samples 45, the mechanical parameters are significantly lower (approximately two times, at most). After the UV process, the strength of the composite remains the same or is higher (Figure 9b). The characteristic parameters of the considered samples are inserted in Table 4 and Table 5. Analyzing the next charts (Figure 10 and Figure 11), in general, the presence of 15 phr of starch in the samples causes a slight decrease in the strength and stiffness, but the addition of 2% wt. of quercetin lowers the mechanical parameters before and after the UV radiation.

## 4. Conclusions

Despite the fact that the addition of 15 phr of starch to the epoxy resin caused a deterioration in the thermal stability, the incorporation of 2 phr of quercetin into the biocomposite resulted in the improvement of this property, and the obtained material filled with a combination of starch and quercetin was characterized by similar thermal properties to the reference sample. Therefore, quercetin may act as a natural thermal stabilizer. Moreover, the composite that contained a combination of potato starch and natural polyphenol exhibited a lower maximum average rate of heat emission (MAHRE) than that of the studied composite containing only starch, and the obtained value was similar to the MAHRE result of the reference sample. The low MAHRE parameter is crucial from a fire safety point of view. The anti-aging effect of quercetin in the biocomposite was detected only on the surface, where the sample containing 15 phr of starch and 2 phr of quercetin was characterized by the lowest carbonyl index value and the lowest polar component of the surface free energy after the solar aging process. Nevertheless, the incorporation of natural additives caused a significant decrease in the maximum stress in the case of both biocomposites with and without quercetin compared to the referential sample. However, it should be noted that both natural substances prevented materials from the post-curing process, which could occur during the solar aging process.

## Figures and Tables

**Figure 1 materials-16-07080-f001:**
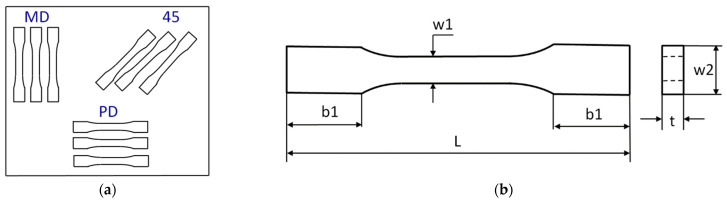
Scheme of taken samples (**a**) and simple drawing (**b**).

**Figure 2 materials-16-07080-f002:**
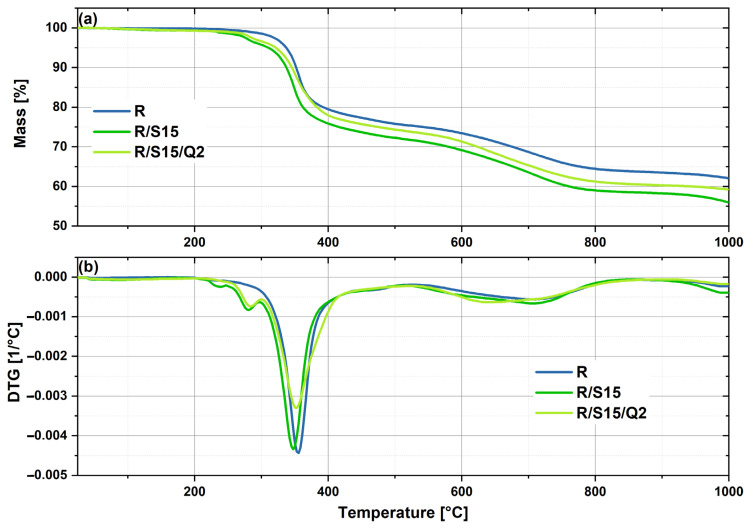
TGA (**a**) and DTG (**b**) curves of the studied epoxy resin-based samples with natural additives (potato starch and quercetin).

**Figure 3 materials-16-07080-f003:**
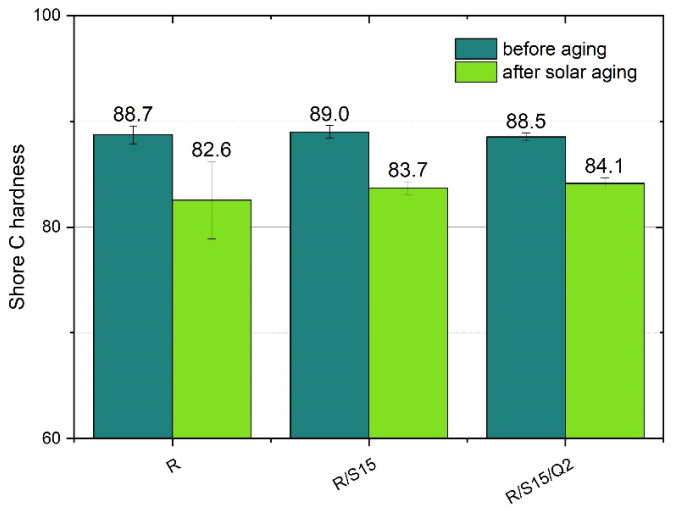
Hardness C results of the studied epoxy resin-based samples with natural additives (potato starch and quercetin).

**Figure 4 materials-16-07080-f004:**
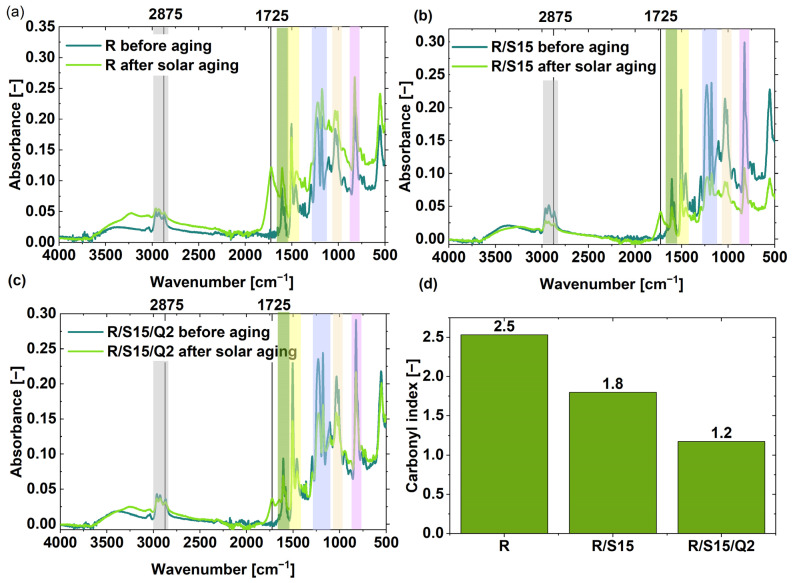
FT-IR spectra of the studied epoxy resin-based samples with natural additives (potato starch and quercetin) before and after solar aging: referential sample (**a**), sample with 15 phr of potato starch (**b**) and sample with 15 phr of potato starch and 2 phr of quercetin (**c**); carbonyl index values of the tested composites (**d**).

**Figure 5 materials-16-07080-f005:**
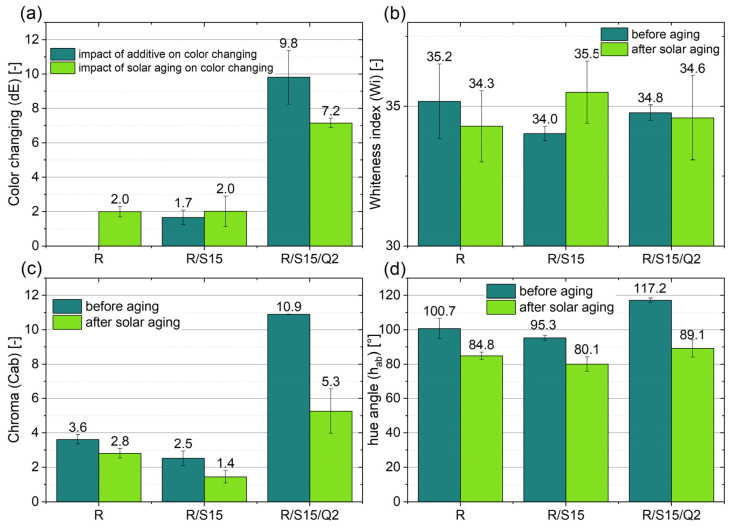
The impact of solar aging on color changing (dE) (**a**), whiteness index (W_i_) (**b**), chroma (C_ab_) (**c**) and hue angle (h_ab_) (**d**) of the studied epoxy resin-based samples with natural additives (potato starch and quercetin).

**Figure 6 materials-16-07080-f006:**
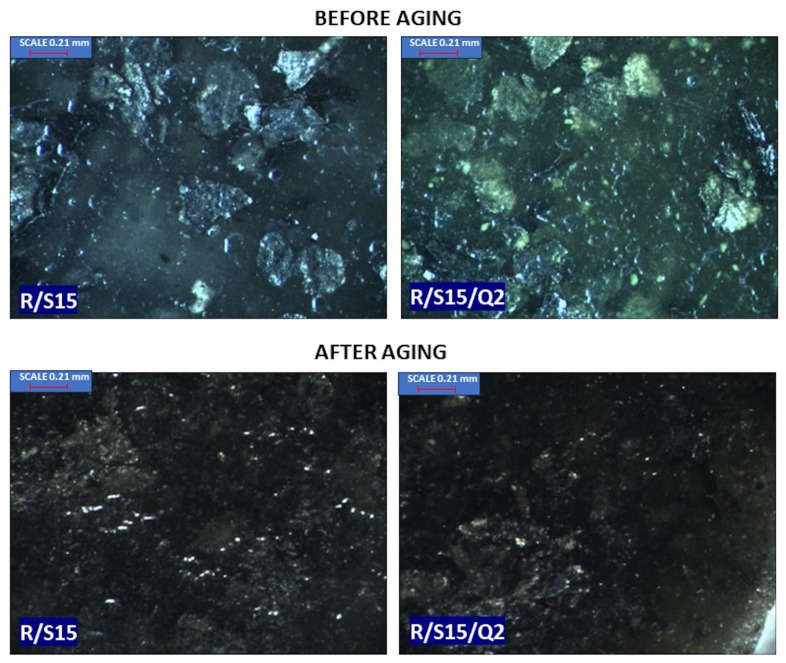
A comparison of the epoxy resin-based sample containing starch with a specimen containing potato starch and quercetin before and after aging.

**Figure 7 materials-16-07080-f007:**
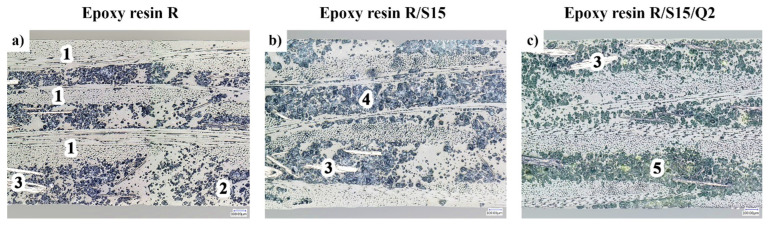
Cross-section images of the epoxy resin composite without additives (R) (**a**), epoxy resin composite containing 15 phr of potato starch (R/S15) (**b**) and epoxy resin composite with 15 phr of potato starch and 2 phr of quercetin (R/S15/Q2) (**c**). All images obtained at a magnification of 200×.

**Figure 8 materials-16-07080-f008:**
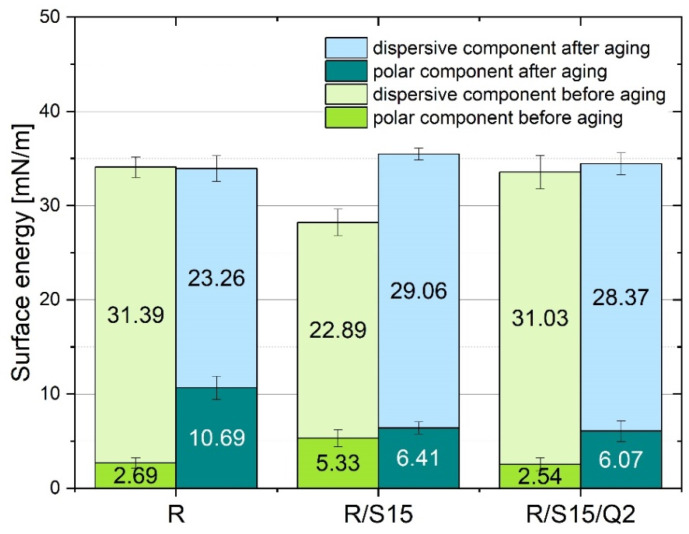
Surface free energy measurement results of the resin composites containing natural additives.

**Figure 9 materials-16-07080-f009:**
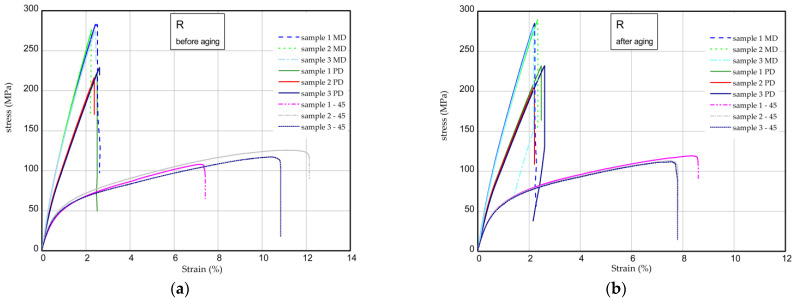
Curves of tension for the pure epoxy resin (R) before aging (**a**) and after aging (**b**).

**Figure 10 materials-16-07080-f010:**
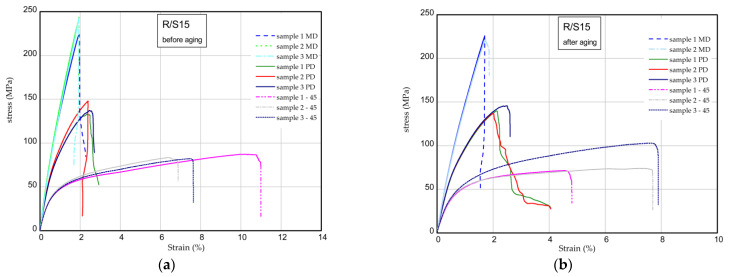
Curves of tension for R/S15 before aging (**a**) and after aging (**b**).

**Figure 11 materials-16-07080-f011:**
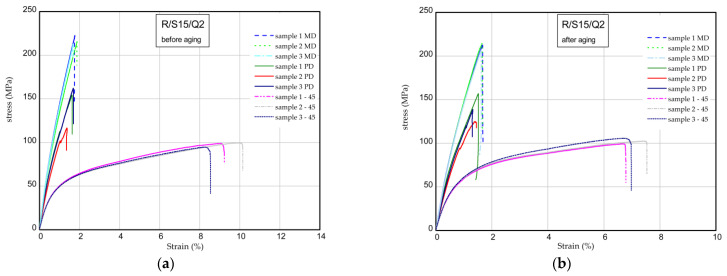
Curves of tension for R/S15/Q2 before aging (**a**) and after aging (**b**).

**Table 1 materials-16-07080-t001:** Composition of the studied epoxy resin-based samples with natural additives.

Sample	Composition	Resin Content [%]	Starch Content [%]	Quercetin Content [%]
R	referential sample (epoxy resin with flame retardants)	100	0	0
R/S15	sample consisted of epoxy resin with flame retardants and 15 phr * of potato starch	87	13	0
R/S15/Q2	sample consisted of epoxy resin with flame retardants and 15 phr * of potato starch and 2 phr * of quercetin	85	13	2

* phr—parts per hundred resin with flame retardants.

**Table 2 materials-16-07080-t002:** Temperature at 5% mass loss (T_05_) [°C], temperature at maximum mass loss rate (T_d_) [°C] and the residual rate of the epoxy resin-based composites with natural additives (potato starch, quercetin) [%].

Sample	T_05_ [°C]	T_d_ [°C]	Residual Rate [%] (T = 1000 °C)
R	327	368.8	62.1
R/S15	310	363.4	56.1
R/S15/Q2	321	367.3	59.3

T_05_—the temperature of 5% material’s mass loss, T_d_—temperature of the most intensive mass loss.

**Table 3 materials-16-07080-t003:** The flammability and smoke emission test results of the epoxy resin-based samples with natural additives.

Sample	MAHRE [kW/m^2^]	D_s_(4) [−]	VOF_4_ [min]
R	48.04	82.69	205.56
R/S15	55.71	127.36	374.36
R/S15/Q2	49.07	144.74	361.16

**Table 4 materials-16-07080-t004:** Main values of Young’ modulus in GPa.

Sample	Before Aging			After Aging		
	MD	PD	45	MD	PD	45
R	17.31 ± 0.20	14.36 ± 0.40	9.34 ± 0.65	18.43 ± 0.24	14.78 ± 0.45	10.39 ± 0.18
R/S15	16.95 ± 0.49	13.51 ± 0.27	8.64 ± 0.17	17.79 ± 0.09	14.33 ± 0.25	9.55 ± 0.35
R/S15/Q2	17.13 ± 0.46	14.01 ± 1.22	8.46 ± 0.20	17.86 ± 0.17	14.69 ± 1.00	9.94 ± 0.17

**Table 5 materials-16-07080-t005:** Main values of maximum stress in MPa.

Sample	Before Aging			After Aging		
	MD	PD	45	MD	PD	45
R	277.7 ± 4.6	223.6 ± 6.7	116.9 ± 8.9	284.9 ± 4.7	223.8 ± 14.2	114.9 ± 4.0
R/S15	236.6 ± 10.9	139.4 ± 7.6	84.3 ± 2.6	223.7 ± 3.1	140.9.3 ± 4.4	82.8 ± 17.5
R/S15/Q2	217.0 ± 5.1	144.4 ± 24.3	97.7 ± 2.6	211.3 ± 4.0	140.2 ± 16.0	102.5 ± 3.2

## Data Availability

The data presented in this study are available in the article.

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
