# Peer review of "Epoxy Resin-Based Materials Containing Natural Additives of Plant Origin Dedicated to Rail Transport"

_materials, 2023, doi:10.3390/ma16227080_

Round 1

Reviewer 1 Report

This paper is concerned with the modification of a commercialized epoxy resin using quercetin hydrate and potato starch. The authors systematacially investigated the effects of the addition of these two bio-based fillers on the solar aging resistance of resulting modified epoxy resins, in terms of hardness, color, mechanical properties, surface free energy and structure variations. The authors also studied the impact of these two bio-based fillers on the thermal stability of resulting modified epoxy resins. In my opinion, this work needs minor revisions before publication in Materials. Specific comments are shown as follows:

(1)  Why didn't the authors investigate the impact of the bio-based fillers on the flame retardancy of the resulting modified epoxy resins, given that they used two bio-based fillers to modify a commercialized epoxy resin with flame retardants.

(2)  What is the specification of glass fabric in this work?

(3)  More discussion on the structure variations based on FTIR results should be added.

The language of this manuscript should be carefully polished. There are grammar errors and non-standard descriptions in English

Author Response

Institute of Polymer and Dye Technology

Lodz University of Technology

90-924 Lodz, ul Stefanowskiego 16, Poland

Tel.: +48 42 631 32 23, Fax: +48 42 636 25 43

October 12, 2023

Materials

Dear Professor,

We are resubmitting our revised paper entitled Epoxy resin-based materials containing natural additives of plant origin dedicated to rail transport by Anna Masek, Olga Olejnik, Leszek Czechowski, Filip Kaźmierczyk, Sebastian Miszczak, Aleksandra Wegier and Sławomir Krauze with a request to reconsider it for publication in Materials.

We have carefully considered the Editor and Reviewers' comments. The manuscript was revised exactly according to these comments. The list of responses to the reviewers’ comments and corrections made in the manuscript is attached.

The manuscript has not been previously published, is not currently submitted for review to any other journal, and will not be submitted elsewhere before a decision is made by this journal.

For correspondence please use the following information:

corresponding author: Anna Masek

Institute of Polymer and Dye Technology

Lodz University of Technology

90-537 Lodz, ul Stefanowskiego 16, Poland

Tel.: +48 42 631 32 93

Fax: +48 42 636 25 43

Yours sincerely,

Ph. D., D.Sc. Anna Masek

Reviewer #1:

Comments and Suggestions for Authors

This paper is concerned with the modification of a commercialized epoxy resin using quercetin hydrate and potato starch. The authors systematacially investigated the effects of the addition of these two bio-based fillers on the solar aging resistance of resulting modified epoxy resins, in terms of hardness, color, mechanical properties, surface free energy and structure variations. The authors also studied the impact of these two bio-based fillers on the thermal stability of resulting modified epoxy resins. In my opinion, this work needs minor revisions before publication in Materials. Specific comments are shown as follows:

  1. Why didn't the authors investigate the impact of the bio-based fillers on the flame retardancy of the resulting modified epoxy resins, given that they used two bio-based fillers to modify a commercialized epoxy resin with flame retardants.

Answer 1 for Reviewer #1:

We appreciate the Reviewer’s attention to this issue. We have added additional information on flammability and smoke emission tests to assess the impact of the added natural ingredients on these parameters:

“3.2. Flammability and smoke emission test

The flammability and smoke emission tests help to assess the potential use of materials in railway seats, which is crucial from the fire safety point of view. One of the most important parameters considered in the design of materials for railway seats is the maximum average rate of heat emission (MAHRE). The tendency of the fire to spread can be measured with this parameter, and according to the EN 45545-2 (R6) standard cannot exceed 90 kW/m2. In spite of the fact that the addition of potato starch into the epoxy resin caused an increase in the MAHRE parameter from 48.04 kW/m2 to 55.71 kW/m2, the obtained value is still lower than 90 kW/m2, which determines the potential use of the designed composite as a material for railway seats. Moreover, the addition of only 2 phr of quercetin to material containing 15 phr of potato starch resulted in a MAHRE value similar to the referential sample and amounted to 49.07 kW/m2.

Another important parameter is Ds(4) parameter, which is the specific optical density in the first 4 minutes of smoke emission during a fire, determined for materials dedicated to rail transport. Based on the EN 45545-2 (R6) standard this parameter cannot exceed 300. The incorporation of 15 phr of potato starch into the epoxy resin caused a significant increase in Ds(4) from 82.69 to 127.36. On the other hand, the composite containing a combination of 15 phr of starch with 2 phr of quercetin was also characterized by a high Ds(4) parameter of 144.74. Nevertheless, these values are still acceptable from the fire safety point of view.

Moreover, the VOF4 parameter, which is the cumulative value of the specific optical densities in the first 4 minutes of the composite fire behavior test, also plays an important role in the design of railway seat materials. This parameter indicates the smoke obscuration in the first 4 minutes of fire and should be less than 600 min. According to the results, all tested materials are characterized by VOF4 parameter below 600 min. The application of 15 phr of starch or a combination of 15 phr of starch with 2 phr of quercetin into epoxy resin composite can change this parameter from 205.56 min to higher values, 374.36 min and 361.16 min  respectively. Nevertheless, the results obtained are not too high and allow the use of the designed composites as railway seat materials.”

Table 3. The flammability and smoke emission test results of epoxy resin-based samples with natural additives.”

Sample

MAHRE [kW/m2]

Ds(4) [-]

VOF4 [min]

R

48.04

82.69

205.56

R/S15

55.71

127.36

374.36

R/S15/Q2

49.07

144.74

361.16

However, it is important to note that potato starch was used as a natural filler and quercetin was added as a natural stabilizer, which was also explained:

“Furthermore, quercetin hydrate (≥95% of purity) obtained from Sigma Aldrich (Munich, Germany) was selected as natural stabilizer and potato starch Melvit® (Kruki, Poland) was applied as filler of plant origin. Both substances were natural additives.”

Reviewer #1:

  1. What is the specification of glass fabric in this work?.

Answer 2 for Reviewer #1:

We thank the Reviewer for this comment. We have decided to fill in the missing details about glass fabric as follows:

The main reinforcing material used to obtain the composites was glass woven roving fabric (2/2, 350 g/m2) produced by Rymatex Sp. z o.o. (Rymanów, Poland).

Reviewer #1:

  1. More discussion on the structure variations based on FTIR results should be added.

Answer 3 for Reviewer #1:

We appreciate the Reviewer’s suggestions. We have added extended discussion about the resin structure as follows:

The aging resistance of epoxy resin-based composites was also investigated using Fourier transform infrared spectroscopy. The structure of the commercial epoxy resin used (NEMresin 1011, a snap cure type epoxy resin) was not revealed by the producer. However, some characteristic moieties were detected using FT-IR spectra. For instance, the triple band visible at 3000-2800 cm-1 (grey highlight) is related to the -CH stretching and the intensity of these bands changes during aging. The intensive peak noticeable at approx. 1606 cm-1 corresponds to the C=C stretching, which is characteristic for aromatic compounds (green highlight). The band at around 1505 cm-1 (yellow highlight) is also related to stretching vibration in the aromatic ring. On the other hand, the significant peak present at about 1231 cm-1 (blue highlight) is due to the ester moieties and/or oxirane rings characteristic for epoxy materials. Furthermore, the double signal visible at 1000-1050 cm-1 comes from -CH2-OH groups (orange highlight). Finally, the intense peak at around 825 cm-1 is due to the presence of C-C stretching vibrations of the epoxy ring (purple highlight). Similar moieties are present in other epoxy resin structures [30]. During the aging process, the new bands at approx. 1725 cm-1 appear as a result of oxidation process, where carbonyl groups are formed. Furthermore, during aging some changes can also occur at around 3430 cm-1, where new -OH groups are formed [32].

Reviewer 2 Report

Growing environmental awareness and concern for sustainability has increased the demand for more environmentally friendly materials. This study focuses on the use of commercially available epoxy resins along with flame retardants, quercetin hydrate and potato starch to create more environmentally friendly materials for rail transportation. I would recommend publication of this manuscript because it provides important insights for the industrial use of epoxy resins. However, the authors need to address the following concerns.

(1) The authors should provide the chemical structures of the epoxy resins studied. For example, a clear structure should be provided, such as in the following reference

https://doi.org/10.1039/D2CP03354B

(2) Regarding a statement " If the carbonyl index is high, it means that many carbonyl groups appeared as a result of oxidation process and the material surface degradates.", it is related to the comment in (1), but the authors should show how carbonyl groups are introduced into the chemical structure of the epoxy resin using the chemical structural formula.

Author Response

Institute of Polymer and Dye Technology

Lodz University of Technology

90-924 Lodz, ul Stefanowskiego 16, Poland

Tel.: +48 42 631 32 23, Fax: +48 42 636 25 43

October 12, 2023

Materials

Dear Professor,

We are resubmitting our revised paper entitled Epoxy resin-based materials containing natural additives of plant origin dedicated to rail transport by Anna Masek, Olga Olejnik, Leszek Czechowski, Filip Kaźmierczyk, Sebastian Miszczak, Aleksandra Wegier and Sławomir Krauze with a request to reconsider it for publication in Materials.

We have carefully considered the Editor and Reviewers' comments. The manuscript was revised exactly in accordance with these comments. The list of responses to the reviewers’ comments and corrections made in the manuscript is attached.

The manuscript has not been previously published, is not currently submitted for review to any other journal, and will not be submitted elsewhere before a decision is made by this journal.

For correspondence please use the following information:

corresponding author: Anna Masek

Institute of Polymer and Dye Technology

Lodz University of Technology

90-537 Lodz, ul Stefanowskiego 16, Poland

Tel.: +48 42 631 32 93

Fax: +48 42 636 25 43

Yours sincerely,

Ph. D., D.Sc. Anna Masek

Reviewer #2:

Comments and Suggestions for Authors

Growing environmental awareness and concern for sustainability has increased the demand for more environmentally friendly materials. This study focuses on the use of commercially available epoxy resins along with flame retardants, quercetin hydrate and potato starch to create more environmentally friendly materials for rail transportation. I would recommend publication of this manuscript because it provides important insights for the industrial use of epoxy resins. However, the authors need to address the following concerns.:

  1. The authors should provide the chemical structures of the epoxy resins studied. For example, a clear structure should be provided, such as in the following reference. https://doi.org/10.1039/D2CP03354B

Answer 1 for Reviewer #2:

We appreciate the Reviewer’s attention to this issue. However, the structure of the commercial epoxy resin used (NEMresin 1011, a snap cure type epoxy resin) was not revealed by the producer. Nevertheless, we have added extended discussion about the resin structure as follows:

The aging resistance of epoxy resin-based composites was also investigated using Fourier transform infrared spectroscopy. The structure of the commercial epoxy resin used (NEMresin 1011, a snap cure type epoxy resin) was not revealed by the producer. However, some characteristic moieties were detected using FT-IR spectra. For instance, the triple band visible at 3000-2800 cm-1 (grey highlight) is related to the -CH stretching and the intensity of these bands changes during aging. The intensive peak noticeable at approx. 1606 cm-1 corresponds to the C=C stretching, which is characteristic for aromatic compounds (green highlight). The band at around 1505 cm-1 (yellow highlight) is also related to stretching vibration in the aromatic ring. On the other hand, the significant peak present at about 1231 cm-1 (blue highlight) is due to the ester moieties and/or oxirane rings characteristic for epoxy materials. Furthermore, the double signal visible at 1000-1050 cm-1 comes from -CH2-OH groups (orange highlight). Finally, the intense peak at around 825 cm-1 is due to the presence of C-C stretching vibrations of the epoxy ring (purple highlight). Similar moieties are present in other epoxy resin structures [30]. During the aging process, the new bands at approx. 1725 cm-1 appear as a result of oxidation process, where carbonyl groups are formed. Furthermore, during aging some changes can also occur at around 3430 cm-1, where new -OH groups are formed [32].

Reviewer #2:

  1. Regarding a statement " If the carbonyl index is high, it means that many carbonyl groups appeared as a result of oxidation process and the material surface degradates.", it is related to the comment in (1), but the authors should show how carbonyl groups are introduced into the chemical structure of the epoxy resin using the chemical structural formula.

Answer 2 for Reviewer #2:

We thank the Reviewer for this comment. As was mentioned, the structure of the commercial epoxy resin used (NEMresin 1011, a snap cure type epoxy resin) was not revealed by the producer. Nevertheless, we have modified the further part of the discussion as follows:

The spectra obtained made it possible to calculate the carbonyl index, which is a good way of assessing the ageing resistance of the materials surface. If the carbonyl index is high, it means that many carbonyl groups have appeared while the amount of -CH moieties has decreased as a result of the oxidation process. This phenomenon is one of the symptoms of material surface degradation. According to Figure 4., the addition of 15 phr of potato starch caused an improvement in the resistance of epoxy resin-based materials to solar ageing, where the carbonyl index decreased from 2.5 to 1.8. The incorporation of 2 phr of quercetin into the R/S15 composite resulted in the lowest carbonyl index parameter, which was 1.2.

Reviewer 3 Report

The combination of natural substances like potato starch and quercetin hydrate with epoxy resin demonstrates a forward-thinking approach to materials science. The comprehensive testing and analysis performed in this study provide strong evidence of the viability and acceptability of these composites in real-world applications. 

1.To further strengthen the study, it is recommended to include an assessment of the impact resistance of the epoxy resin. Evaluating its ability to withstand impact is crucial, especially in applications like rail transport where materials may encounter mechanical stresses.

Author Response

Institute of Polymer and Dye Technology

Lodz University of Technology

90-924 Lodz, ul Stefanowskiego 16, Poland

Tel.: +48 42 631 32 23, Fax: +48 42 636 25 43

October 12, 2023

Materials

Dear Professor,

We are resubmitting our revised paper entitled Epoxy resin-based materials containing natural additives of plant origin dedicated to rail transport by Anna Masek, Olga Olejnik, Leszek Czechowski, Filip Kaźmierczyk, Sebastian Miszczak, Aleksandra Wegier and Sławomir Krauze with a request to reconsider it for publication in Materials.

We have carefully considered the Editor and Reviewers' comments. The manuscript was revised exactly in accordance with these comments. The list of responses to the reviewers’ comments and corrections made in the manuscript is attached.

The manuscript has not been previously published, is not currently submitted for review to any other journal, and will not be submitted elsewhere before a decision is made by this journal.

For correspondence please use the following information:

corresponding author: Anna Masek

Institute of Polymer and Dye Technology

Lodz University of Technology

90-537 Lodz, ul Stefanowskiego 16, Poland

Tel.: +48 42 631 32 93

Fax: +48 42 636 25 43

Yours sincerely,

Ph. D., D.Sc. Anna Masek

Reviewer #3:

The combination of natural substances like potato starch and quercetin hydrate with epoxy resin demonstrates a forward-thinking approach to materials science. The comprehensive testing and analysis performed in this study provide strong evidence of the viability and acceptability of these composites in real-world applications.

  1. To further strengthen the study, it is recommended to include an assessment of the impact resistance of the epoxy resin. Evaluating its ability to withstand impact is crucial, especially in applications like rail transport where materials may encounter mechanical stresses.

Answer 1 for Reviewer #3:

We appreciate the Reviewer’s suggestions. At the moment we are not able to assess the impact resistance of the epoxy resin due to specimens and infrastructure limitations. However, we are aware that the proposed parameter is crucial. Therefore we will pay attention to this aspect in further extended studies.

Reviewer 4 Report

Materials-2610895

The manuscript written by Masek et al., titled “Epoxy resin-based materials containing natural additives of plant origin dedicated to rail transport” provided a study of the use of natural filler and stabilizer to increase the antiaging of commercial epoxy composite with fire retardant.

Please address the following concerns to make it suitable for Materials.

1.      What is the rationale behind adding only 15% of starch and 2% of quercetin additives?

2.      It seems like addition of starch leads to coagulation of the fire-retardant additive, in that case, what is the speculation about the efficient fire retardancy?

3. The addition of quercetin increases the color of the aged product significantly, in that case the authors justify its addition as it is not desirable considering epoxy resins are known for their clear nature.

4.      The retention of the hardness after aging and the decrease in the polar component after aging are similar for the composited with and without quercetin. How this justifies the addition of 2% quercetin which will drive the cost of the formulation.

5.      Please extend the introduction by referring to work that has added antioxidants (synthetic/natural) to decrease the aging degradation of epoxy resin composites. The authors can also point out the fact that natural epoxy resins are also studied to increase sustainability. These references might be helpful from the Webster group. https://doi.org/10.1016/j.polymer.2021.124191, https://doi.org/10.1016/j.porgcoat.2022.106996.

Author Response

Institute of Polymer and Dye Technology

Lodz University of Technology

90-924 Lodz, ul Stefanowskiego 16, Poland

Tel.: +48 42 631 32 23, Fax: +48 42 636 25 43

October 12, 2023

Materials

Dear Professor,

We are resubmitting our revised paper entitled Epoxy resin-based materials containing natural additives of plant origin dedicated to rail transport by Anna Masek, Olga Olejnik, Leszek Czechowski, Filip Kaźmierczyk, Sebastian Miszczak, Aleksandra Wegier and Sławomir Krauze with a request to reconsider it for publication in Materials.

We have carefully considered the Editor and Reviewers' comments. The manuscript was revised exactly in accordance with these comments. The list of responses to the reviewers’ comments and corrections made in the manuscript is attached.

The manuscript has not been previously published, is not currently submitted for review to any other journal, and will not be submitted elsewhere before a decision is made by this journal.

For correspondence please use the following information:

corresponding author: Anna Masek

Institute of Polymer and Dye Technology

Lodz University of Technology

90-537 Lodz, ul Stefanowskiego 16, Poland

Tel.: +48 42 631 32 93

Fax: +48 42 636 25 43

Yours sincerely,

Ph. D., D.Sc. Anna Masek

Reviewer #4:

The manuscript written by Masek et al., titled “Epoxy resin-based materials containing natural additives of plant origin dedicated to rail transport” provided a study of the use of natural filler and stabilizer to increase the antiaging of commercial epoxy composite with fire retardant.

  1. What is the rationale behind adding only 15% of starch and 2% of quercetin additives?.

Answer 1 for Reviewer #4:

We thank the Reviewer for this comment. It is important to note that potato starch was used as a natural filler and quercetin was added as a natural stabilizer, which has been explained below:

Furthermore, quercetin hydrate (≥95% of purity) obtained from Sigma Aldrich (Munich, Germany) was selected as natural stabilizer and potato starch Melvit® (Kruki, Poland) was applied as a filler of plant origin. Both substances were natural additives.”

On the one hand, 15 phr of starch is a sufficient dose of filler that does not significantly affect the properties of the composite and makes the epoxy composite more environmentally friendly. On the other hand, 2 phr of quercetin is a typical and the most effective amount of stabilizer applied to various polymer matrices. This natural substance has been added to prevent premature degradation of the composite during use.

Reviewer #4:

  1. It seems like addition of starch leads to coagulation of the fire-retardant additive, in that case, what is the speculation about the efficient fire retardancy?

Answer 2 for Reviewer #4:

We appreciate the Reviewer’s attention to this issue. We have added additional information about flammability and smoke emission tests to assess the impact of the added natural ingredients on these parameters:

3.2. Flammability and smoke emission test

The flammability and smoke emission tests help to assess the potential use of materials in railway seats, which is crucial from the fire safety point of view. One of the most important parameters considered in the design of materials for railway seats is the maximum average rate of heat emission (MAHRE). The tendency of the fire to spread can be measured with this parameter, and according to the EN 45545-2 (R6) standard cannot exceed 90 kW/m2. In spite of the fact that the addition of potato starch into the epoxy resin caused an increase in the MAHRE parameter from 48.04 kW/m2 to 55.71 kW/m2, the obtained value is still lower than 90 kW/m2, which determines the potential use of the designed composite as a material for railway seats. Moreover, the addition of only 2 phr of quercetin to material containing 15 phr of potato starch resulted in a MAHRE value similar to the referential sample and amounted to 49.07 kW/m2.

Another important parameter is Ds(4) parameter, which is the specific optical density in the first 4 minutes of smoke emission during a fire, determined for materials dedicated to rail transport. Based on the EN 45545-2 (R6) standard this parameter cannot exceed 300. The incorporation of 15 phr of potato starch into the epoxy resin caused a significant increase in Ds(4) from 82.69 to 127.36. On the other hand, the composite containing a combination of 15 phr of starch with 2 phr of quercetin was also characterized by a high Ds(4) parameter of 144.74. Nevertheless, these values are still acceptable from the fire safety point of view.

Moreover, the VOF4 parameter, which is the cumulative value of the specific optical densities in the first 4 minutes of the composite fire behavior test, also plays an important role in the design of railway seat materials. This parameter indicates the smoke obscuration in the first 4 minutes of fire and should be less than 600 min. According to the results, all tested materials are characterized by VOF4 parameter below 600 min. The application of 15 phr of starch or a combination of 15 phr of starch with 2 phr of quercetin into epoxy resin composite can change this parameter from 205.56 min to higher values, including 374.36 min and 361.16 min respectively. Nevertheless, the results obtained are not too high and allow the use of the designed composites as railway seat materials.”

Table 3. The flammability and smoke emission test results of epoxy resin-based samples with natural additives.”

Sample

MAHRE [kW/m2]

Ds(4) [-]

VOF4 [min]

R

48.04

82.69

205.56

R/S15

55.71

127.36

374.36

R/S15/Q2

49.07

144.74

361.16

Reviewer #4:

  1. The addition of quercetin increases the color of the aged product significantly, in that case the authors justify its addition as it is not desirable considering epoxy resins are known for their clear nature.

Answer 3 for Reviewer #4:

We thank the Reviewer for this suggestion. We have added additional discussion in this part of study as follows:

In general, pure epoxy resins are known for their clear nature, but the studied commercially available epoxy resin with flame retardants (NEMresin 1011) is characterized by a dark color, which does not change significantly during aging, even when the material is useless.”

“Additionally, relevant changes in chroma appeared in the R/S15/Q2 sample after solar aging process. Therefore, quercetin can be applied as a kind of natural colorant and also as a natural aging indicator in the dark type of commercial resin, which informs about the material us-age.”

Reviewer #4:

  1. The retention of the hardness after aging and the decrease in the polar component after aging are similar for the composited with and without quercetin. How this justifies the addition of 2% quercetin which will drive the cost of the formulation.

Answer 4 for Reviewer #4:

We are grateful for the Reviewer’s comments. As was mentioned in the article: “The incorporation of 2 phr of quercetin into the biocomposite resulted in the improvement of thermal stability and the obtained material filled with a combination of starch and quercetin was characterized by similar thermal properties to the reference sample. Therefore, quercetin may act as a natural thermal stabilizer.” Moreover, we have already added flammability results, thus it was explained that “(…) the composite containing a combination of potato starch and natural polyphenol exhibited lower maximum average rate of heat emission (MAHRE) than the studied composite containing only starch and the obtained value was similar to the MAHRE result of the reference sample. The low MAHRE parameter is crucial from the fire safety point of view.”

Another aspect is that: “The anti-aging effect of quercetin in the biocomposite was detected only on the surface, where the sample containing 15 phr of starch and 2 phr of quercetin was characterized by the lowest carbonyl index value after solar aging process”.

It means that the application of quercetin provides various benefits in the prepared composites. However, it is also related to extra costs, which is a drawback. Nevertheless, it depends on the producer if the additional advantages will be provided taking into account extra expenses.

Reviewer #4:

  1. Please extend the introduction by referring to work that has added antioxidants (synthetic/natural) to decrease the aging degradation of epoxy resin composites. The authors can also point out the fact that natural epoxy resins are also studied to increase sustainability. These references might be helpful from the Webster group. https://doi.org/10.1016/j.polymer.2021.124191, https://doi.org/10.1016/j.porgcoat.2022.106996.

Answer 5 for Reviewer #4:

We appreciate Reviewer’s suggestions. We have added extra information and cited proposed publications. The extended part of introduction is presented as follows:

“There is great need for novel pro-ecological materials in different sectors, including railway industry. For example, increased attention is being paid to the development of novel biocomposites for seats, which should be lightweight and durable [30]. In modern railways, heavy and expensive metal parts of seats have been replaced by polymer composites, including materials made of epoxy resin, with desirable mechanical properties [31]. However, epoxy resin-based materials, because of their low resistance to aging processes, such as thermal, UV and natural aging [32], require different modifications and the incorporation of various stabilizers. Both natural and synthetic stabilizers dedicated to epoxy resin have been extensively studied [32–34]. However, it must be noted that novel materials should comply with the rules of Green Chemistry. Therefore, many environmentally friendly modification methods of widely used conventional epoxy resin are carried out. On the other hand, the creation of novel fully bio-based composites based on resins derived from renewable resources is also a good way to extend sustainability [35,36]. In this research we are focused on achieving biocomposites dedicated to railway seats by modifying non-sustainable epoxy resin using natural additives, including starch and quercetin.”

Round 2

Reviewer 4 Report

Thank you for addressing the comments.